# Childhood peer status and circulatory disease in adulthood: a prospective cohort study in Stockholm, Sweden

Alexander Miething ![ORCID], Ylva Brännström Almquist

Department of Public Health Sciences, Stockholm University, Stockholm, Sweden

**Correspondence to**
Dr Alexander Miething;
alexander.miething@su.se

## ABSTRACT

**Objectives** Childhood conditions have been recognised as important predictors of short-term and long-term health outcomes, but few studies have considered status position in the peer group as a possible determinant of adult health. Lower peer status, which often reflects experiences of marginalisation and rejection by peers, may impose inequality experiences and leave long-lasting imprints on health. The present study aimed to examine whether peer status is associated with the risk for circulatory disease in adulthood.

**Design** Prospective cohort study based on the Stockholm Birth Cohort Multigenerational Study (SBC Multigen).

**Setting** Stockholm metropolitan area.

**Participants** All individuals who were born in 1953 and resident in the greater metropolitan area of Stockholm in 1963 (n=14 608). The analytical sample consisted of 5410 males and 5990 females. Peer status was sociometrically assessed in cohort members at age 13. The survey material was linked to inpatient care registers that contained information about circulatory diseases (n=1668) across ages 20–63. Cox proportional hazard models were used for the analysis.

**Outcome measure** Circulatory disease.

**Results** Peer marginalisation at age 13 resulted in significantly higher risks of circulatory disease in adulthood among males (HR 1.34; 95% CI 1.09 to 1.64) and females (HR 1.33; 95% CI 1.04 to 1.70) alike. A graded relationship between peer status and circulatory diseases was detected in females (p=0.023). Among males there was a threshold effect, showing that only those in the lowest status position had significantly increased risks of circulatory disease. The associations remained significant after adjusting for various conditions in childhood and adulthood.

**Conclusions** This study shows that circulatory diseases in adulthood may be traceable to low peer status and marginalisation in childhood. It is suggested that peer status in late childhood may precede social integration in adolescence and adulthood, acting as a long-term stressor that contributes to circulatory disease through biological, behavioural and psychosocial pathways.

## INTRODUCTION

Innumerable studies have demonstrated the importance of childhood conditions for later life outcomes and, within health inequality

### Strengths and limitations of this study

► The longitudinal data material with linkages to register data allowed us to study the long-term implications of childhood conditions for health outcomes in adulthood.
► The follow-up period from 1973 to 2016 was relatively long.
► Peer status in childhood as well as other covariates were assessed at a single time point.
► Despite the use of a rich set of covariates, information on certain childhood adversities (eg, child maltreatment and peer victimisation) was unavailable.
► There was little information available about health and health behaviours in childhood and across the subsequent life course that might explain the demonstrated associations.

research, parental socioeconomic conditions are often put forward as key determinants for adult health.[1–3] For example, adverse socioeconomic circumstances in childhood have been shown to predict poor mental health, but also metabolic problems, cardiovascular diseases and higher mortality risks.[4–6] Children nevertheless also hold positions in status structures outside the family, such as in their peer group and school contexts. Yet, relatively few studies have addressed how experiences tied to these childhood-specific types of social structure are related to health outcomes in later life.[7 8] The present study therefore aimed to investigate the long-term implications of the child's own social position—peer status—for the risk of circulatory disease, which represents one of the leading causes of mortality.

Peer orientation significantly increases in early adolescence.[9] Seeking greater autonomy, children distance themselves from their parents and simultaneously intensify their interactions with their peers.[10 11] Accordingly, the role of peer- and school-related factors becomes increasingly important. Intensified peer orientation has also been hypothesised to have buffering effects on

socioeconomic inequalities in health. In conjunction with the 'processes of equalisation' hypothesis, the socioeconomic gradient in health tends to attenuate during the child–youth transition while experiences related to peer interactions and youth culture become more salient.[12] A plausible explanation is that peer interactions and the school environment impose a homogenising effect on children's background characteristics and supersede the repercussions of children's socioeconomic origin. Instead of socioeconomic conditions, the peer context and children's position in the peer group emerge as important determinants of health and for the adoption of health behaviours. Several studies have shown that the initiation of health-related behaviours, such as smoking, is substantially driven by peers rather than determined by socioeconomic conditions.[13–15]

Peers function as socialising agents and are important for children's social, emotional and behavioural development. Interactions with peers facilitate experiences that parents cannot supply,[16] and contribute to identity formation[17] and personality maturation.[18] Furthermore, these interactions have the potential to increase children's self-worth, the ability to build relationships in later life and adaptive adjustment.[19 20] Children are usually well aware of the advantages related to higher status in the peer group.[21] Moreover, their concerns about popularity and reputation significantly increase when they approach adolescence.[22] These affection-based dimensions play an important role in the formation of sociometric peer status, which represents an established instrument to assess the degree to which the child is a liked, accepted and integrated member of the peer group.[7 23] Marginalisation and peer rejection—conditions that are reflected by a low peer status position—deprive children of experiences that are important for their subsequent emotional and cognitive development, and may thus increase the risk for poor well-being and reduced mental health.[24] Östberg and Modin[25] showed a graded inverse relationship between peer status and long-standing illness. Accordingly, although disadvantages arise most clearly for children in the bottom group, they are also tangible for those with higher ranks in the peer hierarchy. However, it is rather obvious that marginalised and socially isolated children suffer the most as they are exceedingly vulnerable to bullying and other forms of destructive peer interactions.[26] Because (pre)adolescence coincides with an increased developmental need and strong desire for peer affiliations, peer rejection and marginalisation may have pervasive impacts on current well-being and health and also affect children's ability to engage in social relationship later on.[21 25 27] As acknowledged through the theory of cumulative disadvantages,[28 29] adverse conditions and negative experiences in childhood tend to accrue across the life course, and magnify detriments in later health as well.[30]

Among adults, stronger social integration has been linked to reduced risks of hypertension, cardiovascular diseases, obesity and mortality,[31–33] indicating that psychosocial mechanisms are at play. It is reasonable to assume that similar processes link children's peer status to later circulatory diseases.[34] Higher status positions in the peer group enable children to derive greater emotional, cognitive and social support from social contacts and facilitate the access to tangible and non-tangible resources provided by social contacts. In line with the buffering hypothesis of social support,[35] supportive social ties impose indirect protective effects on health as they enhance individuals' coping ability and moderate their neuroendocrine or behavioural responses to acute or chronic stress.[36–38] High peer status may also directly enhance individuals' self-care, achievements and health behaviours—factors that might mediate the link to circulatory diseases in adulthood.[39] Confounding factors may nevertheless induce spurious associations between peer status and circulatory disease in later life. Plausible confounding variables are aggressive or controversial behaviours and lower cognitive abilities that are associated with peer rejection as well as poor adult health. Likewise, family-related circumstances such as number of siblings and birth order/sibling position possibly influence children's standing in the peer group, and may also impose long-term effects on health.

Based on data from a Swedish cohort born in 1953 who have been followed for more than 60 years, the present study examines the association between sociometrically assessed peer status in childhood (age 13) and circulatory disease during adulthood (ages 20–63). Drawing on earlier research that has demonstrated socioeconomic inequalities in the risks for cardiovascular and circulatory diseases, acting through psychosocial mechanisms in the adult population, an inverse association between peer status and circulatory diseases is hypothesised. Compared with an earlier study that used the same register-linked survey material,[7] the present study considers a substantially longer follow-up time and a larger set of covariates to examine the influence of third variables.

## METHODS
### Data
The data material used in the current study was the Stockholm Birth Cohort Multigenerational Study (SBC Multigen). In 2018/2019, the SBC Multigen was established through probability matching between two longitudinal anonymised data materials. The first was the Stockholm Metropolitan Study (SMS), defined as all individuals who were born in 1953 and resident in the greater metropolitan area of Stockholm in 1963 (n=15 117). It encompasses survey and register information collected at several time points from 1953 up until 1986, after which data were anonymised. The second was RELINK53, defined as all individuals born in 1953 who lived in Sweden in 1960, 1965 and/or 1968, as well as their ascendant, contemporaneous and descendant family members (n=2 390 753). RELINK53 contains detailed information collected from a variety of administrative registers between

1960 and 2018. An algorithm based on a set of variables present in both data materials rendered it possible to match 14 608 of the SMS cohort members to RELINK53 (for more information, see Almquist et al[40]). Accordingly, we are able to follow these individuals from birth up until their 60s. The Regional Ethical Review Board in Stockholm approved the creation of RELINK53 as well as the probability matching to the SMS that resulted in SBC Multigen (no 2017/34-31/5; 2017/684–32).

### Dependent variable

Information on circulatory diseases was drawn from the Swedish hospital discharge register, covering the years 1973–2016. This register records all inpatient care events (with overnight stays) in Swedish hospitals. In this study, circulatory diseases comprised all diagnoses included in the International Classification of Diseases (ICD 10th revision; chapter IX (I00–I99)). The diagnoses were primarily based on judgements of medical doctors. Preceding versions of the ICD (8th and 9th revision) apply to diagnoses made from 1973 to 1986 and from 1987 to 1996, respectively. These older classifications were harmonised in accordance with the 10th revision of the ICD. The dependent variable operationalised any events of circulatory disease occurring between 1973 and 2016. If multiple events occurred, only the first was taken into account.

### Explanatory variable

Peer status was measured through the question 'Whom do you best like working with at school?'.[7 41] Students were asked to nominate three classmates, in no particular order. This sociometric test was conducted among all grade 6 students in the Stockholm metropolitan area in 1966, with the exception of school classes containing students with learning disabilities. Nominations from classmates who attended grade 6 but were not part of the cohort were also registered. Cohort members' mean (SD) age was 12.9 (0.29) years. For the purpose of the current study we additionally excluded those who attended school classes with fewer than 10 children since the peer status distribution tends to be somewhat different in small classes. Based on the number of received nominations, four categories were created: 'marginalised' denoting 0 nominations; 'low status' 1 nomination; 'medium status' 2 or 3 nominations; and 'high status' ≥4 nominations.

### Covariates

The covariates included family-related, school-related and individual circumstances and conditions. The family-related factors considered the presence of siblings, parents' education and socioeconomic conditions, parents' psychiatric problems and conditions during delivery. Number of siblings and sibling position were assessed in 1972. The latter measured whether the study participant was the only child in the family or whether he/she was last born, middle born or first born. Parental educational level accounted for whether there was at least one adult in the household with an upper secondary

school degree. Parents' receipt of social welfare benefits accounts for socioeconomic difficulties and was available from the social registers, covering the period 1953–1965. Parental psychiatric problems (available from 1953 to 1972) were derived from the social registers and measured whether fathers or mothers showed symptoms of mental illness or psychiatric problems. Information on intervention during the delivery was based on delivery records. Cognitive ability can be regarded as an individual-level confounder that affects a child's position in the peer group but also predicts adverse health behaviours and the risk of poor health in later life. A test of cognitive ability was conducted in 1966 and measured participants' verbal, spatial and numerical competences. Test scores (with a possible range from 0 to 120) were grouped into quartiles of cognitive ability. Average school marks measured study participants' school performance in the sixth grade. Average teacher-reported conduct marks assessed in the sixth grade as well as crimes of violence recorded between 1962 and 1972 indicate participants' externalising problems. Similar to school marks, conduct marks and violent behaviour may act as a third variable in the hypothesised association between peer status and circulatory disease. Mental disorders (ICD-10; chapter V (F00–F09; F2–F6; F70; F99)) and disorders due to alcohol use (ICD 10; chapter V (F10)) manifested in adulthood were drawn from register-based inpatient care data (recorded from 1973 to 2016). Both measures were coded as binary variables and indicate whether a diagnosis was registered at least once between 1973 and 2016.

### Study population

The study population included all cohort members in the SBC Multigen who attended school classes with at least 10 children and had complete information on the variables of interest. Accordingly, the analytical sample resulted in 5410 males and 5990 females.

### Statistical methods

Sex-specific Cox proportional hazard models were used to estimate the risks of circulatory disease. As information on circulatory morbidity was available from 1973 to 2016, study subjects were left truncated at 1 January 1973. Right censoring was applied when the observation ended on 31 December 2016, or earlier, if subjects had died from any disease. Hierarchical regression was chosen as the modelling strategy. Peer status was entered as a single predictor in Model 1. Family-related variables were successively added to Models 2–4. Model 4 additionally included individuals' cognitive ability. Model 5 also controlled for school and conduct marks, as well as crimes of violence. Finally, Model 6 further considers mental problems and disorders due to alcohol use that were diagnosed in adulthood. All analyses were performed with Stata (Version 15.1).

### Patient and public involvement

No patients or members of the public were involved in the study design or conduct of this study.

## RESULTS

The descriptive statistics are shown in table 1 and illustrate a higher prevalence of circulatory diseases for males (18.5%; n=999) than for females (11.2%; n=669). The distribution of peer status is relatively similar across the sexes, although a slightly higher proportion of males (33.1%; n=1788) than females (28.6%; n=1710) were in high status positions, at the same time as more females (15.7%; n=940) than males (12.1%; n=652) experienced marginalisation.

The results from the Cox proportional hazard models are presented in table 2 for males and in table 3 for females. The unadjusted HRs of males' peer status in Model 1 (table 2) show that only those who are in a marginalised position have an increased risk for circulatory disease (HR 1.54; 95% CI 1.27 to 1.87). Controlling for sibling status and number of siblings in Model 2 does not affect the estimates. When adding parental educational level and social welfare receipt to the model, a small decrease in the HR for those in marginalised peer positions can be observed (Model 3). A likelihood ratio test confirmed that the decrease is significant compared with Model 2. The estimates are further attenuated when we adjust for parental psychiatric problems, intervention during delivery and cognitive ability in Model 4. Model 5 additionally considers conduct marks, average school marks and crimes of violence. Although the HR decreases, a significant association between marginalisation and circulatory disease is retained. This association also persists when accounting for mental disorders and disorders due to alcohol misuse in Model 6.

Table 3 presents the corresponding analysis for females. Model 1 illustrates a graded association between peer status and circulatory disease. Females with medium peer status have a 28% higher risk of being hospitalised due to circulatory disease than high status females. The respective increase by low peer status is 42%, and 61% for females in a marginalised position. As shown for males, a gradual attenuation of the HRs of peer status can be observed when control variables are successively added to the model. In Model 4 the HR for medium status is no longer significant whereas, in Model 5, only marginalised females show a significant risk for circulatory disease (HR 1.36; 95% CI 1.07 to 1.74). A similar pattern with a somewhat lowered risk of circulatory disease in marginalised females is shown in Model 6 (HR 1.33; 95 % CI 1.04 to 1.70). Despite the non-significant estimates of medium and low status, a significant trend in the association between peer status and circulatory disease can still be detected (p=0.023). Whereas medium and low status positions of females indicate larger relative risks of circulatory diseases compared with males, marginalised peer status reveals remarkably similar estimates for males and females.

## DISCUSSION

This study shows that peer status in pre-adolescence is associated with circulatory diseases in adulthood. More

**Table 1** Descriptive statistics for the study variables (n=11 400)

| | Males (n=5410) | | Females (n=5990) | |
|---|---|---|---|---|
| | n | % | n | % |
| Circulatory disease (1973–2016) | | | | |
| No | 4411 | 81.5 | 5321 | 88.8 |
| Yes | 999 | 18.5 | 669 | 11.2 |
| Peer status position (1966) | | | | |
| High status | 1788 | 33.1 | 1710 | 28.6 |
| Medium status | 1937 | 35.8 | 2185 | 36.5 |
| Low status | 1033 | 19.1 | 1155 | 19.3 |
| Marginalised | 652 | 12.1 | 940 | 15.7 |
| Sibling position (1972) | | | | |
| Only child | 705 | 13.0 | 705 | 11.8 |
| Last born | 1714 | 31.7 | 1929 | 32.2 |
| Middle born | 1302 | 24.1 | 1509 | 25.2 |
| First born | 1689 | 31.2 | 1847 | 30.8 |
| Number of siblings (1972) | | | | |
| 0 | 705 | 13.0 | 705 | 11.8 |
| 1 | 2090 | 38.6 | 2245 | 37.5 |
| 2 | 1508 | 27.9 | 1675 | 28.0 |
| 3 | 691 | 12.8 | 815 | 13.6 |
| 4 | 254 | 4.7 | 334 | 5.6 |
| More than 4 | 162 | 3.0 | 216 | 3.6 |
| Parental educational level (1960) | | | | |
| At least one parent with upper secondary degree | 1402 | 25.9 | 1533 | 25.6 |
| No parent with upper secondary degree | 3731 | 69.0 | 4131 | 69.0 |
| Missing | 277 | 5.1 | 326 | 5.4 |
| Parental social welfare receipt (1953–1965) | | | | |
| No | 4544 | 84.0 | 5067 | 84.6 |
| Yes | 866 | 16.0 | 923 | 15.4 |
| Parental psychiatric problems (1953–1972) | | | | |
| No | 5105 | 94.4 | 5644 | 94.2 |
| Yes | 305 | 5.6 | 346 | 5.8 |
| Intervention during delivery (1953) | | | | |
| No | 4090 | 75.6 | 4580 | 76.5 |
| Yes | 329 | 6.1 | 293 | 4.9 |
| Missing | 991 | 18.3 | 1117 | 18.7 |
| Cognitive ability (1966) | | | | |
| First quartile (lowest) | 1052 | 19.5 | 1601 | 26.7 |
| Second quartile | 1268 | 23.4 | 1538 | 25.7 |
| Third quartile | 1491 | 27.6 | 1509 | 25.2 |
| Fourth quartile (highest) | 1599 | 29.6 | 1342 | 22.4 |
| Conduct marks (1966) | | | | |
| Not so good | 5 | 0.1 | 1 | 0.0 |
| Good | 83 | 1.5 | 19 | 0.3 |
| Very good | 5322 | 98.4 | 5970 | 99.7 |

Continued

**Table 1** Continued

|  | Males (n=5410) | | Females (n=5990) | |
| --- | --- | --- | --- | --- |
|  | n | % | n | % |
| Average school marks (1966) | | | | |
| Range 1 to 5 | Mean 3.2 | SD 0.9 | Mean 3.3 | SD 0.9 |
| Crimes of violence (1962–1972) | | | | |
| No | 5184 | 95.8 | 5947 | 99.3 |
| Yes | 226 | 4.2 | 43 | 0.7 |
| Mental disorders (1973–2016) | | | | |
| No | 5051 | 93.4 | 5535 | 92.4 |
| Yes | 359 | 6.6 | 455 | 7.6 |
| Disorders due to alcohol use (1973–2016) | | | | |
| No | 5070 | 93.7 | 5804 | 96.9 |
| Yes | 340 | 6.3 | 186 | 3.1 |

specifically, being in a marginalised position in the peer group is significantly associated with increased risks of circulatory disease in both males and females. In addition, a graded association is detected for females: the lower the peer status, the higher the risk of circulatory disease. Compared with the most popular females, even those with a medium-high position in the peer group run higher risks of circulatory diseases in later life. The corresponding analyses of males revealed a threshold effect. Only socially isolated (marginalised) males displayed significantly higher hazards of circulatory disease. The demonstrated associations remained significant also after controlling for variables that could be regarded as potential confounders, including family-related factors and socioeconomic circumstances. Adjustments for school achievements (measured with school marks) and misconduct (measured with conduct marks and crimes of violence) reduced the strength of association to some, but still to a non-significant extent. Mental disorders and disorders due to alcohol misuse in adulthood are likely influenced by adversities during childhood.[42 43] In line with earlier studies, these factors were demonstrated as determinants of circulatory diseases.[44 45] Therefore, the slightly lowered magnitude in the association between peer status and circulatory disease in the female sample points to a weak third variable effect of mental health and alcohol-related problems.

Altogether, our findings suggest an increased susceptibility of marginalised peers for circulatory disease in later life, which is in line with other studies that considered long-term impacts of peer integration.[4 7 31 46] However, this remains to be confirmed with data materials that also encompass additional types of confounding and mediating factors.

Past research has argued that peer status may be considered as a childhood-specific type of social structure that in many ways resembles social status positions in adult life.[7 41] Accordingly, it is plausible that lower peer status would have health implications similar to socioeconomic deprivation and low socioeconomic position in adulthood. As also outlined earlier in this study, peer status may even equalise the effects of low parental socioeconomic position.[12]

The current findings contribute to research on the long-lasting impacts of childhood circumstances on health.[8 47] Studies on health implications of adults' social relationships, however, tend to disregard the emergence of socially-induced health problems and the origin of destructive social experiences at earlier stages in life.[48] In line with the life course theory, deficient social relationships—as well as health problems—may be the result of cumulative processes that escalate with increasing age. Accordingly, earlier research conjectured a mutual interdependence of social relationships and health at any stages in life course.[25] The results found in the current study suggest that this process can be backtracked to adolescence and childhood.

Among adults, stress-buffering effects of tangible and non-tangible support have been proposed as primary mechanisms as to why constructive social relationships are beneficial for health.[35 39 49] Low peer status, social isolation and marginalisation may be particularly distressing in childhood. Socially isolated children suffer from lack of social and emotional support and also from the experience of having reduced opportunities to make friends and to control dominating behaviour by others.[24] Taken together, these adversities might disturb children's social and emotional development and lead to behaviours and disadvantages that are difficult to rectify in their later life. Other important aspects that potentially contribute to the findings in this study are peer victimisation and bullying. Such experiences often coincide with marginalisation and rejection. It is well documented that victimised children and youth have an increased propensity to develop somatic problems, internalising problems, anxiety and depression disorders and tend to take up smoking.[50] Longitudinal studies have shown that such problems persist throughout adulthood, which is reflected by lower mental and physical health.[51 52]

Low peer status in childhood and adolescence has been identified as a source of chronic (psychosocial) stress that furthers the development of mental health problems and chronic inflammation.[4 53 54] In adulthood, the lack of social contacts is associated with disorders that develop over a long time span, including metabolic problems, hypertension, cardiovascular problems and stroke. Earlier research on childhood conditions and life course processes,[55–57] as well as the findings of the current study, suggest that the childhood perspective, on the one hand, and the adulthood perspective, on the other hand, should not be considered as independent given that socially-induced morbidity and mortality in adulthood may have their origin in childhood. There is convincing evidence from neuroscience regarding how social relationships modulate neuroendocrine responses that subsequently affect the circulatory system, increasing the risk for stroke and cardiovascular diseases.[58] Similar mechanisms may

**Table 2** Association between peer status position (1966) and circulatory disease (1973–2016) among males: results from Cox proportional hazard regression models (n=5410)

| | Circulatory disease (1973–2016) | | | | | | | | | | | |
| | Model 1 | | Model 2 | | Model 3 | | Model 4 | | Model 5 | | Model 6 | |
| | HR | 95% CI | HR | 95% CI | HR | 95% CI | HR | 95% CI | HR | 95% CI | HR | 95% CI |
| **Peer status position (1966)** | | | | | | | | | | | | |
| High status (ref) | 1 | | 1 | | 1 | | 1 | | 1 | | 1 | |
| Medium status | 1.13 | (0.97 to 1.32) | 1.12 | (0.96 to 1.31) | 1.11 | (0.95 to 1.29) | 1.08 | (0.92 to 1.26) | 1.06 | (0.90 to 1.24) | 1.05 | (0.89 to 1.23) |
| Low status | 1.14 | (0.95 to 1.37) | 1.13 | (0.94 to 1.36) | 1.11 | (0.93 to 1.33) | 1.07 | (0.89 to 1.28) | 1.03 | (0.85 to 1.24) | 1.01 | (0.84 to 1.22) |
| Marginalised | 1.54*** | (1.27 to 1.87) | 1.53*** | (1.26 to 1.87) | 1.49*** | (1.23 to 1.81) | 1.41*** | (1.16 to 1.72) | 1.34** | (1.10 to 1.64) | 1.34** | (1.09 to 1.64) |
| **Sibling position (1972)** | | | | | | | | | | | | |
| Only child (ref) | | | 1 | | 1 | | 1 | | 1 | | 1 | |
| Last born | | | 0.87 | (0.70 to 1.09) | 0.89 | (0.71 to 1.11) | 0.92 | (0.73 to 1.15) | 0.92 | (0.74 to 1.16) | 0.93 | (0.74 to 1.17) |
| Middle born | | | 0.82 | (0.61 to 1.09) | 0.83 | (0.62 to 1.11) | 0.86 | (0.64 to 1.15) | 0.86 | (0.64 to 1.15) | 0.88 | (0.66 to 1.18) |
| First born | | | 0.85 | (0.67 to 1.06) | 0.86 | (0.69 to 1.09) | 0.87 | (0.70 to 1.10) | 0.88 | (0.70 to 1.11) | 0.89 | (0.71 to 1.12) |
| Number of siblings (1972) | | | 1.06 | (0.99 to 1.13) | 1.04 | (0.97 to 1.11) | 1.03 | (0.96 to 1.10) | 1.03 | (0.96 to 1.10) | 1.02 | (0.95 to 1.09) |
| **Parental educational level (1960)** | | | | | | | | | | | | |
| At least one parent with upper secondary degree (ref) | | | | | 1 | | 1 | | 1 | | 1 | |
| No parent with upper secondary degree | | | | | 1.03 | (0.88 to 1.19) | 0.98 | (0.84 to 1.14) | 0.97 | (0.83 to 1.13) | 0.97 | (0.83 to 1.14) |
| Missing | | | | | 1.00 | (0.73 to 1.35) | 0.90 | (0.65 to 1.25) | 0.90 | (0.65 to 1.24) | 0.91 | (0.66 to 1.25) |
| **Parental social welfare receipt (1953–1965)** | | | | | | | | | | | | |
| No (ref) | | | | | 1 | | 1 | | 1 | | 1 | |
| Yes | | | | | 1.32** | (1.12 to 1.56) | 1.29** | (1.08 to 1.54) | 1.24* | (1.04 to 1.48) | 1.21* | (1.01 to 1.45) |
| **Parental psychiatric problems (1953–1972)** | | | | | | | | | | | | |
| No (ref) | | | | | | | | | | | 1 | |

Continued

**Table 2** Continued

### Circulatory disease (1973–2016)

| | Model 1 | | Model 2 | | Model 3 | | Model 4 | | Model 5 | | Model 6 | |
|---|---|---|---|---|---|---|---|---|---|---|---|---|
| | HR | 95% CI | HR | 95% CI | HR | 95% CI | HR | 95% CI | HR | 95% CI | HR | 95% CI |
| Yes | | | | | | | 1.06 | (0.81 to 1.38) | 1.05 | (0.81 to 1.37) | 1.00 | (0.77 to 1.31) |
| **Intervention during delivery** | | | | | | | | | | | | |
| No (ref) | | | | | | | 1 | | 1 | | 1 | |
| Yes | | | | | | | 1.35* | (1.06 to 1.72) | 1.35* | (1.06 to 1.72) | 1.35* | (1.06 to 1.72) |
| Missing | | | | | | | 1.14 | (0.96 to 1.35) | 1.13 | (0.95 to 1.34) | 1.13 | (0.95 to 1.34) |
| **Cognitive ability (1966)** | | | | | | | | | | | | |
| First quartile (lowest) (ref) | | | | | | | 1 | | 1 | | 1 | |
| Second quartile | | | | | | | 0.90 | (0.75 to 1.07) | 0.91 | (0.76 to 1.09) | 0.94 | (0.78 to 1.13) |
| Third quartile | | | | | | | 0.81* | (0.68 to 0.98) | 0.83* | (0.69 to 1.00) | 0.86 | (0.72 to 1.04) |
| Fourth quartile (highest) | | | | | | | 0.77** | (0.63 to 0.93) | 0.79* | (0.65 to 0.95) | 0.82* | (0.68 to 1.00) |
| Conduct marks (1966) | | | | | | | | | 0.65* | (0.46 to 0.92) | 0.68* | (0.48 to 0.96) |
| Average school marks (1966) | | | | | | | | | 0.97 | (0.90 to 1.03) | 0.96 | (0.90 to 1.03) |
| **Crimes of violence (1962–1972)** | | | | | | | | | | | | |
| No (ref) | | | | | | | | | 1 | | 1 | |
| Yes | | | | | | | | | 1.59** | (1.23 to 2.07) | 1.44*** | (1.10 to 1.87) |
| **Mental disorders (1973–2016)** | | | | | | | | | | | | |
| No (ref) | | | | | | | | | | | 1 | |
| Yes | | | | | | | | | | | 1.43** | (1.14 to 1.79) |
| **Disorders due to alcohol use (1973–2016)** | | | | | | | | | | | | |
| No (ref) | | | | | | | | | | | 1 | |
| Yes | | | | | | | | | | | 1.74*** | (1.39 to 2.17) |

*p<0.05; **p<0.01; ***p<0.001.

**Table 3** Association between peer status position (1966) and circulatory disease (1973–2016) among females: results from Cox proportional hazard regression models (n=5990)

| | Circulatory disease (1973–2016) | | | | | | | | | | | |
|---|---|---|---|---|---|---|---|---|---|---|---|---|
| | Model 1 | | Model 2 | | Model 3 | | Model 4 | | Model 5 | | Model 6 | |
| | HR | 95% CI | HR | 95% CI | HR | 95% CI | HR | 95% CI | HR | 95% CI | HR | 95% CI |
| **Peer status position (1966)** | | | | | | | | | | | | |
| High status (ref) | 1 | | 1 | | 1 | | 1 | | 1 | | 1 | |
| Medium status | 1.28* | (1.05 to 1.57) | 1.27* | (1.03 to 1.55) | 1.24* | (1.01 to 1.51) | 1.20 | (0.98 to 1.48) | 1.18 | (0.96 to 1.45) | 1.18 | (0.96 to 1.45) |
| Low status | 1.42** | (1.13 to 1.79) | 1.40** | (1.11 to 1.76) | 1.36** | (1.08 to 1.71) | 1.30* | (1.03 to 1.64) | 1.26 | (0.99 to 1.59) | 1.24 | (0.98 to 1.57) |
| Marginalised | 1.61*** | (1.27 to 2.03) | 1.54*** | (1.22 to 1.95) | 1.49*** | (1.18 to 1.89) | 1.42** | (1.11 to 1.81) | 1.36* | (1.07 to 1.74) | 1.33* | (1.04 to 1.70) |
| **Sibling position (1972)** | | | | | | | | | | | | |
| Only child (ref) | | | 1 | | 1 | | 1 | | 1 | | 1 | |
| Last born | | | 0.76* | (0.58 to 1.00) | 0.81 | (0.62 to 1.07) | 0.82 | (0.62 to 1.07) | 0.82 | (0.63 to 1.08) | 0.83 | (0.63 to 1.10) |
| Middle born | | | 0.66* | (0.47 to 0.92) | 0.69* | (0.50 to 0.97) | 0.68* | (0.49 to 0.96) | 0.69* | (0.49 to 0.97) | 0.70* | (0.50 to 0.98) |
| First born | | | 0.64** | (0.48 to 0.84) | 0.68** | (0.51 to 0.90) | 0.69** | (0.52 to 0.91) | 0.69* | (0.53 to 0.92) | 0.70* | (0.53 to 0.93) |
| Number of siblings (1972) | | | 1.12*** | (1.05 to 1.19) | 1.10** | (1.03 to 1.17) | 1.10** | (1.03 to 1.17) | 1.10** | (1.03 to 1.18) | 1.10** | (1.03 to 1.18) |
| **Parental educational level (1960)** | | | | | | | | | | | | |
| At least one parent with upper secondary degree (ref) | | | | | 1 | | 1 | | 1 | | 1 | |
| No parent with upper secondary degree | | | | | 1.53*** | (1.25 to 1.88) | 1.49*** | (1.21 to 1.84) | 1.48*** | (1.20 to 1.82) | 1.48** | (1.20 to 1.82) |
| Missing | | | | | 1.57* | (1.10 to 2.25) | 1.55* | (1.05 to 2.27) | 1.54* | (1.05 to 2.26) | 1.52* | (1.03 to 2.22) |
| **Parental social welfare receipt (1953–1965)** | | | | | | | | | | | | |
| No (ref) | | | | | 1 | | 1 | | 1 | | 1 | |
| Yes | | | | | 1.19 | (0.97 to 1.45) | 0.99 | (0.79 to 1.24) | 0.99 | (0.79 to 1.23) | 0.97 | (0.77 to 1.21) |
| **Parental psychiatric problems (1953–1972)** | | | | | | | | | | | | |
| No (ref) | | | | | | | 1 | | 1 | | 1 | |
| Yes | | | | | | | 1.74*** | (1.31 to 2.30) | 1.74*** | (1.32 to 2.31) | 1.68** | (1.27 to 2.23) |

Continued

**Table 3** Continued

### Circulatory disease (1973–2016)

| | Model 1 | | Model 2 | | Model 3 | | Model 4 | | Model 5 | | Model 6 | |
|---|---|---|---|---|---|---|---|---|---|---|---|---|
| | HR | 95% CI | HR | 95% CI | HR | 95% CI | HR | 95% CI | HR | 95% CI | HR | 95% CI |
| Intervention during delivery | | | | | | | | | | | | |
| No (ref) | | | | | | | 1 | | 1 | | 1 | |
| Yes | | | | | | | 1.26 | (0.90 to 1.76) | 1.25 | (0.89 to 1.74) | 1.24 | (0.89 to 1.73) |
| Missing | | | | | | | 0.98 | (0.79 to 1.22) | 0.98 | (0.79 to 1.21) | 0.97 | (0.78 to 1.21) |
| Cognitive ability (1966) | | | | | | | | | | | | |
| First quartile (lowest) (ref) | | | | | | | 1 | | 1 | | 1 | |
| Second quartile | | | | | | | 0.88 | (0.72 to 1.08) | 0.89 | (0.72 to 1.09) | 0.91 | (0.74 to 1.12) |
| Third quartile | | | | | | | 0.95 | (0.77 to 1.18) | 0.97 | (0.78 to 1.20) | 1.00 | (0.81 to 1.24) |
| Fourth quartile (highest) | | | | | | | 0.82 | (0.64 to 1.04) | 0.84 | (0.66 to 1.06) | 0.87 | (0.68 to 1.11) |
| Conduct marks (1966) | | | | | | | | | 4.56 | (0.66 to 31.66) | 4.01 | (0.58 to 27.70) |
| Average school marks (1966) | | | | | | | | | 0.90* | (0.83 to 0.98) | 0.90* | (0.83 to 0.98) |
| Crimes of violence (1962–1972) | | | | | | | | | | | | |
| No (ref) | | | | | | | | | 1 | | 1 | |
| Yes | | | | | | | | | 2.20* | (1.16 to 4.14) | 1.86 | (0.98 to 3.53) |
| Mental disorders (1973–2016) | | | | | | | | | | | | |
| No (ref) | | | | | | | | | | | 1 | |
| Yes | | | | | | | | | | | 1.38* | (1.07 to 1.77) |
| Disorders due to alcohol use (1973–2016) | | | | | | | | | | | | |
| No (ref) | | | | | | | | | | | 1 | |
| Yes | | | | | | | | | | | 1.94*** | (1.40 to 2.69) |

*p<0.05; **p<0.01; ***p<0.001.

apply to the findings in the current study and explain why marginalisation in the peer group poses a risk factor for the development of circulatory diseases.

Another finding worth noting is the observed differences between males and females. Similar to our results, previous studies have detected a dose–response relationship between females' peer status and metabolic syndrome in adulthood.[46] The graded associations between peer status and circulatory disease in females could suggest a greater emotional and social reactivity to chronic stress in comparison to males.[59 60] The greater vulnerability of girls to interpersonal stress may further be reflected by gender differences in social interactions and acquisition of social status in the peer group. Whereas girls tend to be relationship-oriented, boys are more involved in competitive and organised play.[59] Boys are also more concerned about achievements and problems directly affecting themselves.[61] These differences become manifest in girls' greater worries and self-blaming about others' relationship problems and concerns about negative evaluation by other peers. Because girls pay more attention to their peer environment and social interactions[62], they may react more strongly to feedback from peers. In contrast, boys' stress pertains to school performance, physical competition and close friendships rather than to their social networks. According to Oldehinkel *et al* [24], likeability and affection can be considered as the primary criteria of high peer status in girls, while admiration (of their achievements) marks the status position of boys. However, popular (high-status) peers are not necessarily well liked because peer status may also be achieved by dominance and aggression.[63] Another finding from the sex-specific analysis is the distinct association of crimes of violence with risks of circulatory diseases in males and females. The descriptive analysis clearly indicated that crimes of violence occur less frequently in females than in males, which is in line with the notion that females tend to use more indirect forms of aggression.[64] The relatively high risk for circulatory disease of females involved in violent crimes may further reflect that those females are likely to experience serious mental health problems[65] which, in turn, pose a risk factor for developing circulatory diseases in later life. In fact, the somewhat lower HRs of females' crimes of violence after controlling for mental health problems indicate that both conditions are linked to each other.

### Strengths and limitations

A particular strength of this study is the use of longitudinal data material with linkages to register data. This allowed us to study the long-term implications of childhood conditions for health outcomes in adulthood. Participation rates were high compared with many other studies, which reduced the effects of selection and biases from attrition and non-response. The studied single cohort covered a fairly homogeneous population. Therefore, certain spurious effects, such as influences on the presented associations due to immigration, can be ruled out. Nevertheless, this study is also subject to limitations. Peer status in childhood as well as other covariates were assessed at a single time point. Further, there was little information available about health and health behaviours in childhood and across the subsequent life course (ie, in adolescence and adulthood) that might affect the demonstrated associations. For example, possible third variable effects of child maltreatment, peer victimisation, smoking and substance abuse could not be addressed in the present study. Moreover, it was not possible to determine when health problems (eg, the predecessors of circulatory diseases) emerged. Likewise, potentially relevant information about circumstances in adulthood (eg, achievements and ambitions) was not available in the data. It was therefore not possible to fully identify the causal pathways—for instance, whether psychological, behavioural or biological processes contributed to demonstrated associations between peer status and circulatory disease. Despite the use of controls for parental socioeconomic status and other confounders such as family circumstances, it is nevertheless possible that selection and unmeasured confounding affected the identified associations. Given these limitations, we cannot draw any causal inference from this study.

### CONCLUSIONS

This study confirmed that childhood circumstances impose health effects in later life. Peer relations play an important role for children's emotional and social development and may have considerable long-term implications on their health. Congruent with earlier studies on the role of socioeconomic conditions for morbidity and mortality, the findings of this study stress the importance of children's social relationships for circulatory diseases. Our investigation of this rather specific outcome might help to understand the processes that contribute to socially-induced morbidity and mortality.

**Contributors** AM and YBA conceived this study. YBA prepared and compiled the data. AM performed the statistical analyses. AM and YBA wrote the manuscript. Both authors approved the final version of the manuscript.

**Funding** This work was supported by the Swedish Council for Health, Working Life and Social Research [grant no. 2016–07148], (http://www.forte.se/en/).

**Competing interests** None declared.

**Patient and public involvement** Patients and/or the public were not involved in the design, or conduct, or reporting, or dissemination plans of this research.

**Patient consent for publication** Not required.

**Ethics approval** The study was approved by the Stockholm Regional Ethics Committee (reg. no. 2016/481-31/5; 2016/481-31/5). The need for consent was waived due to the retrospective nature of this study.

**Provenance and peer review** Not commissioned; externally peer reviewed.

**Data availability statement** Data are available upon reasonable request. The datasets generated and/or analysed during the current study are not publicly available due to ethical regulations regarding the Stockholm Birth Cohort (SBC) study but are available from the co-author on reasonable request.

**ORCID iD**

Alexander Miething http://orcid.org/0000-0003-2004-3780

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
