## [Reviewer comments · BMJ Open]

ARTICLE DETAILS

TITLE (PROVISIONAL)	Childhood peer status and circulatory disease in adulthood: a prospective cohort study in Stockholm, Sweden
AUTHORS	Miething, Alexander; Almquist, Ylva

VERSION 1 – REVIEW

REVIEWER	Vera Clemens University of Ulm, Germany
REVIEW RETURNED	04-Feb-2020

GENERAL COMMENTS	The authors assessed the association between peer relation during adolescence and somatic health during adulthood in an epidemiological approach using data from the Stockholm Birth Cohort Multigenerational Study. The manuscript gives an interesting insight into the relevance of peer status on later health. The strengths of the analysis are the perspective character and the linkage of high quality data registers. Nevertheless, there are some aspects that the authors should revise: • My major concern is that it is not stated clearly what is the new in this study. The authors published a study regarding peer status and health (including (eg, ischaemic heart disease and several other health outcomes) in 2009 – please point out what is new regarding this paper• The authors name the high response as a strengths of the study. Even though I agree that the data set is a strength due to the quality and prospective character, I disagree that the response rate is impressively “high” (11,500 of 15,000)• For the family-related factors, no maltreatment was assessed – which is known to be associated with social isolation, bullying and cardiovascular disease in adulthood – if you do not have this data, this should be at least be addressed in the limitations section• Another major confounder is cardiovascular disease of the parents - which is known to be related to low socioeconomic status, child maltreatment and social isolation – and can affect the risk of children via genetical and behavioral (role model) pathways – is there any possibility to include this into the analyses as confounder? Another such potential confounder is mental health of the participants (during childhood and adulthood)• Discussion: “According to Oldehinkel (2007), likability and affection can be considered as the primary criteria of high peer status in girls, while admiration (of their achievements) marks the status position of boys. With regard to our findings, it can be concluded that not belonging to the most liked (i.e., the most popular peers) may hurt more than not belonging to the most admired.” ◊ I do not see how this can be concluded by your data, please revise this paragraph
--

	 • The limitation section starts with strengths – potentially rename the section into strengths and limitations? • The cited literature is quite old, my first impression by screening pubmed is that there are a lot of recent papers that could be added into the discussion.
--	---

REVIEWER	Kirsty Lee University of Warwick
REVIEW RETURNED	13-Feb-2020

GENERAL COMMENTS	Thank you for the opportunity to review this paper on childhood peer status and circulatory disease. Overall this is a nicely written paper that uses data from a large birth cohort and presents some interesting findings. I have a few minor comments/queries, mostly related to the analysis and discussion: Page 7, line 9 – Important confounders missing from the analysis are abuse/neglect at home, peer victimization and smoking/alcohol/drug abuse and mental illness in adulthood. Do you have any data on these? Page 7, line 37 – “information was collected continuously from ...” does this mean at yearly intervals? Page 8, line 22 – what was the mean age (and SD) of the students in grade 6? Page 8, line 14 – You control for lower cognitive abilities, yet students with learning disabilities were excluded from the sociometric assessment. Could these students still be nominated? Were the quintiles reported for students with scores in the “normal” range of cognitive function? Page 9, line 2 – Were the average conduct marks reported by teachers? Page 9, line 19 – more information is needed on the analysis. Circularity disease was operationalised as one recorded event that a required an inpatient stay any time between 1973-2016? How were the models (1-5) built up? What statistical software was used? Page 10, line 45 – crimes of violence among women are more abnormal (girls and women tend to use more indirect forms of aggression) and tend to be associated with more serious mental health problems. The health gap in people with serious mental illness is well documented and is another confounding factor, especially as for girls, parental psychiatric problems were a strong predictor of circulatory problems. This could have been drawn out more in the discussion. Page 11, line 10 – you discuss the role of mediators, but you did not test for mediation in the analysis so I think this is misleading. I also think it is a stretch to suggest there may be an independent association between peer status and circulatory disease. Page 11, line 59 – you touch on the topic of bullying victimisation, but I think this warrants further discussion. Youth rejected by their peers are also often targets of victimisation and much research shows the negative effects of peer victimisation on mental and physical health outcomes. Those rejected and victimised may incur more significant costs to their health - see Knack et al. (2012) What Protects Rejected Adolescents From Also Being Bullied by Their Peers? Page 12, line 58 – Youth can be high in peer status (popular), but not necessarily well-liked, because their status is achieved by dominance and aggression – see Vaillancourt & Hymel (2006) Aggression and Social Status: The Moderating Roles of Sex and Peer-Valued Characteristics.
---

	Page 13, line 20 – I think it is important to point out specifically the confounders that were not adjusted for (abuse/neglect at home, peer victimization and smoking/alcohol/drug abuse and mental illness in adulthood). There were several grammatical errors in the manuscript. It would be worth re-reading and checking this carefully.
--	--

VERSION 1 – AUTHOR RESPONSE

Reviewer: 1

Reviewer Name: Vera Clemens

Institution and Country: University of Ulm, Germany Please state any competing interests or state 'None declared': None

Please leave your comments for the authors below The authors assessed the association between peer relation during adolescence and somatic health during adulthood in an epidemiological approach using data from the Stockholm Birth Cohort Multigenerational Study. The manuscript gives an interesting insight into the relevance of peer status on later health. The strengths of the analysis are the perspective character and the linkage of high quality data registers. Nevertheless, there are some aspects that the authors should revise:

- My major concern is that it is not stated clearly what is the new in this study. The authors published a study regarding peer status and health (including (eg, ischaemic heart disease and several other health outcomes) in 2009 – please point out what is new regarding this paper

Thank you for making us aware of this lack of clarity. Indeed, the study by Almquist (2009) used a similar setup and individuals from the same cohort as in the present study. However, the present study utilized a cohort update, which resulted in a notably longer follow-up time from 1973 to 2016 (instead of 1973-2003). Therefore, we were able to deal with substantially higher number of circulatory disease events as compared to the study from 2009. In the current study, we also included a richer set of control variables which were not part of the study from 2009 (e.g., sibling position, cognitive ability, school and conduct marks).

We added a sentence about the originality of the present study.

p.7: "In comparison to an earlier study that used the same register-linked survey material (Almquist, 2009), the present study considers a substantially longer follow-up time and a larger set of covariates to examine the role of potentially confounding and mediating factors."

More details about the birth cohort and the cohort update are documented in Almquist et al 2019.

(Almquist, Y.B., Grotta, A., Vågerö, D., Stenberg, S.-Å., Modin, B., 2019. Cohort profile update: The Stockholm birth cohort study. *Int. J. Epidemiol.* <https://doi.org/10.1093/ije/dyz185>)

- The authors name the high response as a strengths of the study. Even though I agree that the data set is a strength due to the quality and prospective character, I disagree that the response rate is impressively "high" (11,500 of 15,000)

We agree that this (whether the response rates are high or low) can be disputed. The response rates in many contemporary large-scale survey materials are much lower than in our study. However, we revised our formulation (now stating "high in comparison to many other studies", page 14) and dropped the statement from the list of "Strengths and limitations of this study" (page 3).

- For the family-related factors, no maltreatment was assessed – which is known to be associated with social isolation, bullying and cardiovascular disease in adulthood – if you do not have this data, this should be at least be addressed in the limitations section

Unfortunately, information on maltreatment was unavailable. This is rather unsurprising because such information is highly sensitive. It is now mentioned that this information is unavailable:

p.13: “Further, there was little information available about health and health behaviours in childhood and across the subsequent life course (i.e., in adolescence and adulthood) that might affect the demonstrated associations. For example, possible confounding effects of child maltreatment and peer victimisation, or mediating effects of smoking and substance abuse, could not be addressed in the present study.”

- Another major confounder is cardiovascular disease of the parents - which is known to be related to low socioeconomic status, child maltreatment and social isolation – and can affect the risk of children via genetical and behavioral (role model) pathways – is there any possibility to include this into the analyses as confounder? Another such potential confounder is mental health of the participants (during childhood and adulthood)

Information on cardiovascular disease of the parents was not available. The same applies to specific information about participants’ mental health during childhood. However, we had access to participants’ “mental disorders” (and “disorders due to alcohol use”) during adulthood and included these indicators in our revised analysis (Model 6). Although “mental disorders” and “disorders due to alcohol use” turned out as strong predictors for circulatory disease, they did not substantially affect the association between peer status position and circulatory disease. We revised the results and discussion section accordingly.

- Discussion: “According to Oldehinkel (2007), likability and affection can be considered as the primary criteria of high peer status in girls, while admiration (of their achievements) marks the status position of boys. With regard to our findings, it can be concluded that not belonging to the most liked (i.e., the most popular peers) may hurt more than not belonging to the most admired.” ◊ I do not see how this can be concluded by your data, please revise this paragraph

We agree that our formulation and interpretation was somewhat awkward and dropped this sentence. Please note that the entire paragraph was revised in accordance with the comments from Reviewer #2.

- The limitation section starts with strengths – potentially rename the section into strengths and limitations?

We renamed the section “Strengths and limitations”.

- The cited literature is quite old, my first impression by screening pubmed is that there are a lot of recent papers that could be added into the discussion

Some of the references are indeed somewhat old, but not necessarily outdated because they represent canonical work. Nevertheless, we added some more recent references to the revised parts of the discussion section (e.g. p.11/12: Hughes et al., 2017; Shin et al., 2018; Shonkoff et al., 2012; Rueger et al., 2016).

Reviewer: 2

Reviewer Name: Kirsty Lee

Institution and Country: University of Warwick Please state any competing interests or state 'None declared': None declared

Please leave your comments for the authors below Thank you for the opportunity to review this paper on childhood peer status and circulatory disease. Overall this is a nicely written paper that uses data from a large birth cohort and presents some interesting findings. I have a few minor comments/queries, mostly related to the analysis and discussion:

Page 7, line 9 – Important confounders missing from the analysis are abuse/neglect at home, peer victimization and smoking/alcohol/drug abuse and mental illness in adulthood. Do you have any data on these?

Thank you for this suggestion. Unfortunately, we do not have information about abuse/ neglect at home and peer victimization. The information on smoking is incomplete and therefore not suitable for our analysis. The inpatient-care data, however, included information about “mental disorders” and “disorders due to alcohol use” in adulthood (1973 to 2016).

In the updated analysis (with the newly created Model 6), we now control for mental disorders and disorders due to alcohol use. We revised the methods, results, and discussions accordingly.

Page 7, line 37 – “information was collected continuously from ...” does this mean at yearly intervals?

The used data were collected at different stages and time points depending on the type of data. Information from medical registers was collected yearly as for example events of circulatory diseases, mental disorders, and disorders due to alcohol use. The survey data was collected at a single time point (in 1966). Information about respondents' parents refers various registers and different time points or periods.

We revised our formulation (p.7: “at several time points”).

Page 8, line 22 – what was the mean age (and SD) of the students in grade 6?

The mean age in grade 6 (at the time of the interview) was 12.9 years (SD 0.29). This is now mentioned on page 8.

Page 8, line 14 – You control for lower cognitive abilities, yet students with learning disabilities were excluded from the sociometric assessment. Could these students still be nominated? Were the quintiles reported for students with scores in the “normal” range of cognitive function?

School classes with students that had learning disabilities did not participate in the sociometric test (or in the School Study of 1966). This is now clarified in the methods section:

p.8: “This sociometric test was conducted among all 6-grade students in the Stockholm metropolitan area in 1966, with the exception of school classes containing students with learning disabilities.”

Accordingly, test scores of cognitive function considered the “normal” range.

Page 9, line 2 – Were the average conduct marks reported by teachers?

School and conduct marks were reported by teachers and registered in the local school registers. Conduct marks were assessed in 1966 (and not 1996 as wrongly stated in the previous versions of Table 2 and 3).

p.8: "Average teacher-reported conduct marks assessed in the 6th grade and crimes of violence recorded between 1962 and 1972 indicate participants' externalising problems."

Page 9, line 19 – more information is needed on the analysis. Circulatory disease was operationalised as one recorded event that a required an inpatient stay any time between 1973-2016? How were the models (1-5) built up? What statistical software was used?

The (binary) variable "circulatory disease" recorded any events of inpatient stay due to circulatory disease between 1973 and 2016. Multiple events were coded as one event. Please see our revision/clarification on page 8:

"The using variable operationalised any events of circulatory disease occurring between 1973 and 2016. If multiple events occurred, only the first was taken into account."

We used 'hierarchical modelling' as analytical strategy. Starting with a simple bivariate model, we added childhood conditions (Model 2) followed by (socioeconomic) parental factors (Model 3) and rather cognitive-biological/ medical conditions to Model 4. School and behavioral variables were added to Model 5. Mental- alcohol-related disorders in adulthood are included in Model 6. The analysis was performed with Stata (version 15.1). Our clarification can be found on page 9 in the manuscript.

Page 10, line 45 – crimes of violence among women are more abnormal (girls and women tend to use more indirect forms of aggression) and tend to be associated with more serious mental health problems. The health gap in people with serious mental illness is well documented and is another confounding factor, especially as for girls, parental psychiatric problems were a strong predictor of circulatory problems. This could have been drawn out more in the discussion.

Thank you for this comment. We revised and extended the discussion in accordance with your suggestion.

p.13: "Another finding from the gender-specific analysis is the distinct association of crimes of violence with risks of circulatory diseases in males and females. The descriptive analysis clearly indicated that females' crimes of violence occur less frequent in comparison to males, which is in line with the notion that females tend to use more indirect forms of aggression (Salmivalli and Kaukiainen, 2004). The relatively high risk for circulatory disease of females involved in violent crimes may not least reflect that those females are likely to experience serious mental health problems (Cauffman et al., 2007) that in turn pose a risk factor for developing circulatory diseases in later life. In fact, the somewhat lower hazard ratios of females' crimes of violence after controlling for mental health problems indicate that both conditions are linked to each other."

The revised analysis now accounts for mental problems in adulthood (Model 6).

Page 11, line 10 – you discuss the role of mediators, but you did not test for mediation in the analysis so I think this is misleading. I also think it is a stretch to suggest there may be an independent association between peer status and circulatory disease.

We agree that we should be more cautious with stating mediation. School marks, conduct marks, and crimes of violence could indeed affect peer status and therefore impose confounding effects too. We chose a more careful phrasing in our revision:

p.11: "Adjustments for school achievements (measured with school marks) and misconduct (measured with conduct marks and crimes of violence) reduced the strength of association to some, but still a non-significant extent."

We also agree that we cannot be sure about an "independent association" between peer status and circulatory disease. We revised accordingly:

p.11: "Altogether, our findings suggest an increased susceptibility of marginalised peers for circulatory disease in later life, which is in line with other studies that considered long-term impacts of peer integration"

Page 11, line 59 – you touch on the topic of bullying victimisation, but I think this warrants further discussion. Youth rejected by their peers are also often targets of victimisation and much research shows the negative effects of peer victimisation on mental and physical health outcomes. Those rejected and victimised may incur more significant costs to their health - see Knack et al. (2012) What Protects Rejected Adolescents From Also Being Bullied by Their Peers?

Thank you for pointing this out and providing the reference. We revised the paragraph and added some sentences. We do not cite Knack et al (2012), but added some other references that support our reasoning.

p.12: "Other important aspects that potentially contribute to the findings in this study are peer victimisation and bullying. Such experiences often coincide with marginalisation and rejection. It is well documented that victimised children and youth have an increased propensity to develop somatic problems, internalising problems, anxiety and depression disorders, and tend to take up smoking (Wolke and Lereya, 2015). Longitudinal studies have demonstrated that such problems persist throughout adulthood, which is reflected by lower mental and physical health (Copeland et al., 2014; Lereya et al., 2015)."

Page 12, line 58 – Youth can be high in peer status (popular), but not necessarily well-liked, because their status is achieved by dominance and aggression – see Vaillancourt & Hymel (2006) Aggression and Social Status: The Moderating Roles of Sex and Peer-Valued Characteristics.

Thank you again for this important perspective. We revised the paragraph on page 14 and now refer to Vaillancourt & Hymel (2006).

p.13: "However, popular (high-status) peers are not necessarily well-liked, because peer status may also be achieved by dominance and aggression (Vaillancourt and Hymel, 2006)."

Page 13, line 20 – I think it is important to point out specifically the confounders that were not adjusted for (abuse/neglect at home, peer victimization and smoking/alcohol/drug abuse and mental illness in adulthood).

We agree. It is now stated that important confounders are missing (unavailable) in the study.

p.13: “For example, possible confounding effects of child maltreatment and peer victimisation, or mediating effects of smoking and substance abuse, could not be addressed in the present study.”

There were several grammatical errors in the manuscript. It would be worth re-reading and checking this carefully.

Thank you for making us aware of this. We identified and corrected several grammatical errors.

VERSION 2 – REVIEW

REVIEWER	Vera Ciemens University of Ulm, Germany
REVIEW RETURNED	28-Mar-2020

GENERAL COMMENTS	The authors adressed the reviewer suggestions adequately. Study and results are interesting and I recommend to publish this paper.
--

REVIEWER	Kirsty Lee University of Warwick
REVIEW RETURNED	02-Apr-2020

GENERAL COMMENTS	Thank you for revising the paper. Although you removed the comment about mediating effects as I suggested in the previous review, you included it in a further two places in this revision. I do not think it is appropriate to use the term mediation when you did not test this statistically. My final comment is that you included the term "gender-specific..." but this should be "sex-specific". You measure biological sex and use the terms male and female throughout the document.
---

VERSION 2 – AUTHOR RESPONSE

Authors comments to Reviewer 2

Although you removed the comment about mediating effects as I suggested in the previous review, you included it in a further two places in this revision. I do not think it is appropriate to use the term mediation when you did not test this statistically. My final comment is that you included the term "gender-specific..." but this should be "sex-specific". You measure biological sex and use the terms male and female throughout the document.

We would like to thank you again for your comments. We revised the misleading formulations of ‘mediating effects’. Wherever appropriate, we now use the term ‘third variable effects’ instead. Please see our revisions on page 3 (line 6), page 7 (line 2), page 8 (line 31), page 11 (line 6), and page 13 (lines 18/19). (Please note that page and line numbers refer to the clean document).

We also agree that the term 'gender' should be replaced by 'sex'. We conducted revisions on page 9 (lines 8, 21) and page 12 (line 33).